# Resonant Oscillations of Ion-Stabilized Nanobubbles in Water as a Possible Source of Electromagnetic Radiation in the Gigahertz Range

**DOI:** 10.3390/ijms26146811

**Published:** 2025-07-16

**Authors:** Nikolai F. Bunkin, Yulia V. Novakovskaya, Rostislav Y. Gerasimov, Barry W. Ninham, Sergey A. Tarasov, Natalia N. Rodionova, German O. Stepanov

**Affiliations:** 1Department of Fundamental Sciences, Bauman Moscow State Technical University, 105005 Moscow, Russia; r_guerassimov@mail.ru; 2Chemistry Department, Lomonosov Moscow State University, 119991 Moscow, Russia; 3Materials Physics (Formerly Department of Applied Mathematics), Research School of Physics, Australian National University, Canberra, ACT 2600, Australia; barry.ninham@anu.edu.au; 4Research and Development Department, OOO “NPF “Materia Medica Holding”, 129272 Moscow, Russia or satarasovmail@yandex.ru (S.A.T.); rodionovann@materiamedica.ru (N.N.R.); or stepanov_go@rsmu.ru (G.O.S.); 5Department of Molecular and Cellular Pathophysiology, Institute of General Pathology and Pathophysiology, 125315 Moscow, Russia; 6Institute of Biomedicine, Pirogov Russian National Research Medical University, Ministry of Health of the Russian Federation, 117997 Moscow, Russia

**Keywords:** gas nanobubbles, oscillations of nanobubbles, electromagnetic wave, gigahertz range, dynamic light scattering, vibrational impact

## Abstract

It is well known that aqueous solutions can emit electromagnetic waves in the radio frequency range. However, the physical nature of this process is not yet fully understood. In this work, the possible role of gas nanobubbles formed in the bulk liquid is considered. We develop a theoretical model based on the concept of gas bubbles stabilized by ions, or “bubstons”. The role of bicarbonate and hydronium ions in the formation and stabilization of bubstons is explained through the use of quantum chemical simulations. A new model of oscillating bubstons, which takes into account the double electric layer formed around their gas core, is proposed. Theoretical estimates of the frequencies and intensities of oscillations of such compound species are obtained. It was determined that oscillations of negatively charged bubstons can occur in the GHz frequency range, and should be accompanied by the emission of electromagnetic waves. To validate the theoretical assumptions, we used dynamic light scattering (DLS) and showed that, after subjecting aqueous solutions to vigorous shaking with a force of 4 or 8 N (kg·m/s^2^) and a frequency of 4–5 Hz, the volume number density of bubstons increased by about two orders of magnitude. Radiometric measurements in the frequency range of 50 MHz to 3.5 GHz revealed an increase in the intensity of radiation emitted by water samples upon the vibrational treatment. It is argued that, according to our new theoretical model, this radiation can be caused by oscillating bubstons.

## 1. Introduction

In modern biophysics, the effects of radiation emitted by biological macromolecules in aqueous solutions are an actively researched area. Usually, the physical nature of such radiation is related to chemiluminescence—a chemical oxidation reaction that leads to the emission of light [1]. In this case, the generation of active forms of oxygen in liquid samples is assumed to play a major role [2,3,4]. When biological objects act as a source of chemiluminescence, such luminescence is referred to as bioluminescence [5,6,7], a phenomenon that has been known for a long time [8]. Apparently, one of the very first examples of bioluminescence is the so-called mitogenetic radiation, which was discovered nearly a century ago [9,10,11]. An extremely weak ultraviolet radiation emitted by one biological species (the inducer) can affect the rate of mitosis in another biological species (the detector). The frequency range of luminescence of biological macromolecules extends from the radio range to the near ultraviolet [7]. Some studies have reported non-classical properties of radiation emitted by biological objects (e.g., the effects of “squeezed” light [12]), as well as the coherent nature of such radiation [13]. In addition, spontaneous chemiluminescence of aqueous solutions can be affected by vibrational treatment [14].

However, recent studies have shown that chemiluminescence is not the only source of electromagnetic radiation of biological objects. For example, dilute aqueous solutions of some bacterial and viral DNA under certain conditions can emit low-frequency electromagnetic signals by non-chemiluminescent mechanisms [15]. The authors of this work suggested that such radiation is due to the presence of coherent domains composed of water molecules; the sizes of such domains significantly exceed intermolecular distances. It should be noted that vibrational treatment of liquid samples [16,17], as well as treatment with weak magnetic fields [18,19], also affects such non-chemiluminescent emission. Let us also note that vigorous shaking should inevitably affect the mutual arrangement and, hence, interactions of water molecules both among themselves and with foreign particles, in particular with protein macromolecules.

It is well known that all physical objects in a state of thermodynamic equilibrium absorb and emit electromagnetic waves. The spectral density of this radiation is determined only by temperature. The probability of energy absorption/emission during this process is described by the so-called Einstein coefficients. At the same time, mechanical treatment of water, such as shaking, produces non-monotonic changes in pressure inside the sample (areas of increased and decreased density of the substance), which is inevitably related to significant distortions and the emergence of numerous defects in the hydrogen-bond (H-bond) network. These defects facilitate the dissolution of atmospheric gases in water.

Water and aqueous solutions under normal conditions, i.e., at atmospheric pressure and room temperature, are saturated with atmospheric gases. Gases are ignored in classical theories of physical chemistry: a significant omission. The gases (nitrogen, oxygen), dissolved in water, just like defects or isotopes in solids, attract and self-assemble into dynamic nanobubbles much like surfactant micelles. The dissolved gases keep their molecular structure and form nanosized defects in the H-bond network; these defects can be called nanobubbles of dissolved gas. The presence of nanobubbles is revealed in conductivity measurements as a function of salt concentration. This dependence shows a sharp break at a “critical nanobubble concentration” that coincides with the physiological salt concentration of the blood. It reflects the sudden formation of stable nanobubbles that produce activated gas species for enzymatic reactions [20,21]. One half of all ion pairs exhibit the phenomenon, whereas the other half do not. The same phenomenon occurs with sugars and amino acids. Thus, any attempt to explain this by electrostatic forces will necessarily be incomplete [22,23]. If dissolved gas is removed, no break in the conductivity is observed with changing salt concentration [24,25].

The presence of such nanobubbles was confirmed in experiments on acoustic cavitation, which requires a pressure of about several bars [26], while the theoretical estimate of tensile strength of liquid is 10^4^ bar [27]. In addition, when water is saturated with dissolved gas at a high pressure, and this pressure is sharply dropped, the water–gas solution becomes supersaturated. This also results in the generation of gas bubbles [28]. However, both cavitation and the generation of large gas bubbles under supersaturation conditions require long-lived nuclei, which can be nanobubbles that contain dissolved gas.

It should be noted that when carbon dioxide dissolves in water, bicarbonate and hydronium ions are formed: H_2_O + CO_2_ ⇔ (H_3_O^+^)_aq_ + (HCO_3_^−^)_aq_,(1)

The pKa of the reaction (6.3) is quite high. At the same time, hydronium cations, along with hydroxide anions, are present in water due to the autoprotolysis reaction. These ions have rather high mobility, namely 36.23 × 10^−8^ and 20.64 × 10^−8^ m^2^/(s × V) at 298 K, respectively [29,30]. Shaking a suspension of bacteriophages leads to their inactivation at the liquid–gas interface [17]. It is also known that bubbling CO_2_ through water causes its sterilization due to the increase in the concentration of carboxyl and bicarbonate ions [31].

Bicarbonate ion, as a foreign ion, which is present in water in fairly large quantities (up to 10^−5.5^ mol/L in a wide range of pH values from 6 to 10 [31]), due to its large size, is characterized by significantly lower mobility (4.61 × 10^−8^ m^2^/(s × V)) and high affinity to interphase boundaries [32].

Thus, when solving the problem of electromagnetic radiation sources in water and aqueous solutions, considering their dependence on the structuring of the hydrogen-bond network of water, it seems reasonable to analyze:the state and probable role of nanobubbles, which are defects in the hydrogen-bond network and contain atmospheric gases, including carbon dioxide, the hydration of which produces bicarbonate ions;the nature and concentration of the sources of electromagnetic radiation depending on the vibrational treatment; andthe ability of aqueous solutions to emit electromagnetic radiation upon shaking.

In summary, this work is devoted to the study of physical mechanisms of electromagnetic wave emission from liquid samples, and these mechanisms are not associated with bioluminescence, i.e., with the emission of biological macromolecules due to chemiluminescence, which is excited, as a rule, in the presence of reactive oxygen species. The emission of electromagnetic waves in the model under consideration is caused by oscillations of electrically charged gas nanobubbles at their eigen-frequency. Such oscillations are accompanied by the emission of an electromagnetic wave.

## 2. Results

### 2.1. Experimental Results

In water and aqueous solutions, even in the equilibrium state, gas nanobubbles are present. This can be confirmed by dynamic light scattering (DLS) [33]. Figure 1 shows the size distributions of gas nanobubbles in water and 10 mg/L NaCl solution, which were subjected to shaking at a frequency of 4 Hz and a force of 4 or 8 N (kg·m/s^2^).

As follows from this figure, with an increase in the force applied, the effective size of gas nanobubbles increases: The position of the distribution maximum shifts along the abscissa axis towards the larger sizes. At the same time, after vigorous shaking, the number of nanobubbles also increases, amounting to 10^9^–10^10^ cm^−3^. Note that, as shown in [34], the volume number density of nanobubbles in an equilibrium NaCl solution, which was not subjected to any mechanical treatment such as shaking or stirring, is equal to 5 × 10^7^ cm^−3^. Thus, vigorous shaking of aqueous solutions results in an increase in the volume number density of nanobubbles and their sizes, which was shown for the first time in [34].

The experiments described below were motivated by previously obtained results showing that spontaneous radiation from aqueous solutions of alkali metal chlorides in the radio frequency range does not necessarily relate to equilibrium thermal radiation [35,36]. For analyzing the emission of electromagnetic waves in the radio range, the aqueous samples were subjected to multiple successive hundred-fold dilutions. The dilution process was accompanied by intense mechanical shaking with a frequency of 4 Hz and a force of 4 or 8 N (kg·m/s^2^) (see Section 4.2 for more details). Figure 2 shows the results of measurements of the radiation intensity from water samples in the range of 0.05–3.5 GHz.

As follows from this figure, in aqueous solutions subjected to repeated intensive shaking during successive hundred-fold dilutions, the intensity of electromagnetic emission in the GHz range is approximately twice as high compared to water (reference), which was not subjected to shaking and multiple dilutions. This result is statistically significant (*p* < 0.05). Considering the results shown in Figure 1 and Figure 2, it is reasonable to assume that radiation in the radio frequency range can be attributed to nanobubbles, the volume number density of which increased as a result of shaking; see above. To understand whether and how the nanobubbles can generate electromagnetic waves, it is necessary to have a reliable model of the species. This, in turn, requires both phenomenological macroscopic data about the conditions of their existence and microscopic data about their state and possible structure of boundary layers. The latter can be retrieved from quantum chemical simulations of model systems.

### 2.2. Quantum Chemical Modeling: The State of Hydronium and Bicarbonate Ions in the Nanobubble Shells

The local neighborhoods of molecules in the bulk water or aqueous solution and in the boundary layer of a nanobubble are essentially different. In general, the coordination of a molecule within the shell of a nanobubble in a liquid is close to that of a molecule in the surface layer in contact with the atmosphere. The larger the bubble size, i.e., the smaller the curvature of the phase boundary surface, the closer the forces exerted on the molecule and the mutual arrangement of the neighboring molecules to those typical of nearly flat surface layers. The larger the curvature of the surface layer of a nanobubble, the more substantial the forces (primarily of an electrostatic and dispersion nature) that cause the formation of a three-dimensional H-bond network of water molecules. These forces can lead to a local contraction of nanobubble surface areas due to the convergence and partial reorientation of the water molecules and, in the limiting case, cause the sealing of a cavity in the bond network of molecules. This process is facilitated by the fact that all molecules in the surface layer of a nanobubble are under-coordinated compared to molecules in the bulk liquid. The number of their nearest neighbors is, on average, smaller by one. In the bulk, the optimal coordination of water molecules is tetrahedral. Each molecule forms two hydrogen bonds with the neighbors as a proton donor (*d*; each proton is involved in a coordination bond with the oxygen atom of the neighboring molecule) and two bonds as an acceptor (*a*; the oxygen atom, which has electron density uninvolved in the formation of intramolecular bonds, can coordinate protons of two neighboring water molecules).

In the surface layer, *ddaa* coordination is impossible, and the *dda* and *daa* variants are prevailing. In the former case, both protons are involved in hydrogen bonds. Therefore, the molecule, being only a potential donor of the electron density of the oxygen atom, can additionally coordinate any electron-deficient particle, for example, a cation. In the latter case, the oxygen atom has a locally tetrahedral coordination. One of the protons, which is not involved in any hydrogen bond, is a potentially active coordinating site for electron-excess particles such as foreign anions. Such a proton tends also to form a hydrogen bond with an under-coordinated oxygen atom of a water molecule. This process causes the above-described cross-linking of the fragments of nanobubble walls and its eventual collapse. Thus, one can expect a competition between the processes initiated by the existence of OH groups, which are not involved in the formation of hydrogen bonds, in water molecules inside the boundary layer. If the curvature of the nanobubble surface is small, the coordination of such OH groups at the side of the liquid phase seems preferable, especially if the liquid contains anions.

Let us analyze the structures of aquacomplexes of the bicarbonate anion from this point of view. To provide the local arrangements of molecules in the model clusters that can be considered as similar to those in nanobubble boundary layers, we have carried out preliminary test simulations. We have optimized the structures of individual water clusters, the molecules in which were initially randomly distributed over the sphere surfaces of a gradually increasing radius. The number of molecules was selected in such a way that the final H-bonded network can envelop the whole sphere. The initial radii considered fell in the range of 3 to 7 Å, with the largest number of molecules being 60. An example of the initial cluster composed of 23 water molecules arranged over a sphere of 4 Å radius is shown in Figure 3a. During the optimization procedure, no restriction was imposed. Many resulting structures were characterized by continuous 3D H-bond networks with small cavities inside. Only a few configurations could be classified as hollow clusters. The best variant of this kind composed of 23 molecules is shown in Figure 3b. In the case of the large numbers of molecules (50 to 60), ball-like clusters were found (such as the one shown in Figure 4a). Then, O_2_ or N_2_ molecules were placed inside the hollow clusters, and the resulting complex systems were again optimized. The above 23-molecular cluster was found to be the smallest one in which the gas molecules can be kept, with the vertical dissociation energy of the systems being close to zero. The energy was estimated taking into account the basis set superposition error as follows:*D_e_* = ½ (E^i^(X_2_) + E^i^((H_2_O)_n_)) + ½ (E^f^(X_2_) + E^f^((H_2_O)_n_)) − E((X_2_(H_2_O)_n_),
where E((X_2_(H_2_O)_n_) is the total energy of the cluster at its optimized configuration, and E^i^(F) and E^f^(F) stand for the energies of the corresponding fragment (F = X_2_ or (H_2_O)_n_ and X = O or N) taken at its configuration within the cluster and calculated either with the use of the individual basis set of the fragment (E^i^) or with the use of the full basis set of the whole cluster (E^f^).

At smaller number of molecules, the *D_e_* values in the case of N_2_ and O_2_ molecules were negative. It is worth noting that when these molecules were placed in the center of the cavity formed by 23 water molecules and the whole system relaxed, the cavity shape changed (Figure 3c,d). It became less distorted and more ball-like. This is reflected in both the surface areas of the clusters and their volumes estimated in terms of the overlapping atomic van der Waals spheres (r(H) = 1.2 Å and r(O) = 1.5 Å):
**(H_2_O)_23_****O_2_(H_2_O)_23_****N_2_(H_2_O)_23_**Surface area, Å^2^551597601Volume, Å^3^383407410

Note that the formal volume per one gas molecule in these clusters is smaller than that in the gas phase under normal conditions by a factor of nearly 200, which corresponds to the actual pressures inside nanobubbles that appear under extreme conditions. Under normal conditions, the pressure should be much lower, which is the case for cavities formed by larger numbers of water molecules (about 60). However, diatomic gas molecules contained in H-bonded hollow clusters determine their less defective and more ball-like shapes, with no OH groups uninvolved in hydrogen bonds inside the cavity. This is close to the actual structures of nanobubbles. Therefore, water clusters of this kind, composed of larger numbers of molecules, were selected for subsequent analysis of the role of CO_2_ molecules, whose appearance in water should lead to the formation of bicarbonate and hydronium ions. The state of the ions within the boundary layers of nanobubbles is of primary interest.

Bicarbonate anion [HCO_3_]^−^, surrounded by water molecules, can be viewed in a rough approximation as a combination of OH and CO_2_ fragments, where the OH fragment is almost neutral, having a local charge of only about −0.1–−0.05 a.u., and the CO_2_ fragment is noticeably charged (−0.60 ÷ −0.65 a.u.). Accordingly, the coordinating properties of the anion as a whole are determined by its constituent parts, one of which is close to the ·OH radical and the other to the anion. In full accordance with this model, the OH fragment forms one coordination bond in water clusters as a proton donor, while the CO_2_ fragment, as an electron density donor, forms at least four bonds with water molecules (each oxygen atom coordinates two to three molecules).

Figure 4 shows two optimized structures of the [HCO_3_(H_2_O)_60_]^−^ cluster anion, which correspond to local minima of the adiabatic potential. In both cases, a stable spherical cluster containing 60 water molecules was used as an initial approximation (Figure 4a), and the bicarbonate anion was located near its surface, either on the outside or on the inside. At the initial location of the HCO_3_ fragment outside the cluster, a structure shown in Figure 4b is formed. Here, the cluster segment (which involves 14 water molecules closest to HCO_3_^−^) is significantly deformed: being initially convex, it becomes almost flat. This reflects the fact that the coordinating ability of the HCO_3_ fragment is quite high: the potential gradient it creates is sufficient for the convergence and reorientation of water molecules, which provides for the formation of the maximum possible number of hydrogen bonds between the HCO_3_ fragment and water molecules.

At the initial location of the same fragment on the inner side of the spherical water cluster, i.e., in the region of the local concavity of the hydrogen-bond network, its position turns out to be more stable, and the reorganization of the cluster is more substantial (Figure 4c). In fact, an internal molecular compartment arises in the cluster, and the HCO_3_ fragment plays the role of a reinforcing element in it, directly coordinating six water molecules strictly in its first solvation shell. Such a configuration is energetically more favorable: the difference in adiabatic potentials is −30.1 kcal/mol, and the difference in Gibbs vibrational energies is −23.5 kcal/mol. Such a large difference is due, among other things, to the existence of a more complete first solvation shell of the anion. This reflects its inclination to be embedded inside a layer of water molecules rather than being localized on its surface. Thus, the location of the bicarbonate anion on the inner side of the bubble–water interface actually causes the local contraction of the nanobubble walls, leading to a decrease in its volume. Finally, the bicarbonate anion finds itself in an almost completely formed first solvation shell. This may be the result of the hydration of a carbon dioxide molecule enclosed in a nanobubble inside water.

It is noteworthy that, in both cases, the HCO_3_ fragment is either embedded in a nearly flat (according to the arrangement of oxygen atoms) substructure of water molecules or coordinated near it. Moreover, if the total number and mutual arrangement of water molecules are sufficient and suitable, hydrogen bonds formed by each oxygen atom of the CO_2_ subfragment form a pyramidal sector with an apex angle of at least 40°. At the same time, the water molecules that form hydrogen bonds with this subfragment, due to their own flat structure, have very different orientations in space, without preferential closure of their OH groups within small local domains. This makes the nature of H-bonding of water molecules different from that typical of water surface layers, reducing the energy of their interaction with each other, i.e., reducing the surface tension.

The inclination of a bicarbonate ion to be built into the H-bond network of water molecules is even more clearly manifested in the case when the ion is initially located in a spherical layer of molecules (Figure 5). This is well illustrated even when the optimal configuration of the system (corresponding to the minimization of energy gradients) is found for the system in terms of the stationary approach. Note that this approach does not take into account any thermal perturbation of the system, which significantly expands the possibilities of energy redistribution during the actual dynamic reorganization of the cluster. Nevertheless, even in the absence of the actual thermal excitation of the system, the inclusion of a bicarbonate anion in a monolayer of water molecules leads to the appearance of such noticeable forces that a significant part of the cluster undergoes reorganization. As follows from Figure 5, during the reorganization process, the HCO_3_ fragment rotates, and its initially tangential orientation with respect to the spherical cluster (Figure 5a) turns into the normal orientation (Figure 5b). This is accompanied by the formation of hydrogen bonds with water molecules, which are very similar in their configuration to those in the above-considered variant when the bicarbonate ion was initially placed inside a spherical cluster of water molecules. About 20 molecules, i.e., a third of the cluster, are directly involved in this process. In the corresponding [HCO_3_(H_2_O)_20_]^−^ subsystem, a three-dimensional network of hydrogen bonds, typical of the bulk phase, is formed, and the initially 60-molecular spherical cavity is reduced to a 40-molecular one.

This can be considered as additional evidence that a stable variant of localization of the bicarbonate ion in water can be a subsurface location close to the phase boundary, but inside at least a bi- or trimolecular (in thickness) layer. In this case, the orientation of the ion with respect to the spherical boundary of the nanobubble can be either tangential or normal, and about 90% of its total charge (about −0.7 a.u.) is concentrated on the CO_2_ fragment.

It should also be noted that when a bicarbonate ion is localized in a water cluster, its flat structure and the presence of a carbon atom, which cannot be integrated into the hydrogen-bond network of water molecules, promote a local distortion of the H-bond network. This is well illustrated by the structures of the [HCO_3_(H_2_O)_39_]^−^ cluster (Figure 6). Subnanometer-sized cavities arise above and below the plane where the nuclei of this ion are located. Parts of the boundaries of these subnanocavities are six- and seven-membered rings of water molecules. This peculiarity is independent of whether the bicarbonate ion is localized inside a cluster of water molecules (Figure 6a) or on its surface (Figure 6b). Here, the total defectiveness of the H-bond network turns out to be expectedly greater in the case of internal localization of the ion in the cluster: the number of hydrogen bonds formed in the system is 69 and 74 in the cases of internal and superficial location of the ion. Nevertheless, its internal location turns out to be energetically more preferable. The difference in adiabatic potentials of the structures shown in Figure 6a,b is −11.9 kcal/mol, and the thermal vibrational contributions to the Gibbs energy additionally increase the energy gap to 13.2 kcal/mol.

Thus, the disturbance introduced by the bicarbonate ion into the H-bond network of water is compensated by the formation of coordination bonds with water molecules. The distortions of the network itself, which are manifested in a decrease in the local density of molecules at both sides of the CO_3_ plane at a concurrent increase in the density around the oxygen nuclei, actually make the ion the reinforcing block within the layer of water molecules, whose oxygen atoms are located in the same plane as the oxygen atoms of the ion. At the same time, two adjacent layers of water molecules, which are necessary for the energetically stable localization of the ion, are located at a distance larger than that typical of the H-bond network of water molecules in the bulk phase.

When carbon dioxide dissolves in water (see reaction (1)), hydrated hydronium ions are formed. These are water ions and therefore easily build into the hydrogen-bond network. Each hydronium ion forms strong H-bonds with three water molecules as a proton donor of hydrogen bonds, so that a stable, long-lived H_9_O_4_^+^ fragment is formed. The oxygen atom of this ion manifests a very weak tendency to participate in hydrogen bonds as a proton acceptor (donor of electron density) due to its small negative charge (about −0.25 a.u. compared to −0.5 or −0.6 a.u. in water molecules). Accordingly, in water clusters, either the H_3_O^+^ ion is localized in the surface layer (Figure 7a) or, at a side of its oxygen atom in the H-bond network, a seven-membered H-bonded ring is formed, whereas four-, five-, or six-membered rings are typical of water (Figure 7b). The surface location of this ion in a small cluster of water molecules is more energetically favorable. The difference in the adiabatic energies of the structures shown in Figure 7 is −8.4 kcal/mol, being only slightly smaller when the thermal vibrational excitation is taken into account (−7.8 kcal/mol). In general, this means that the hydronium ion can also be embedded in the surface layer of water molecules surrounding the gas core of a nanobubble. In this case, it will promote the orientation of the OH groups of the water molecules closest to it in a direction opposite to the gas core. This should generally reduce the number of “unbound” OH groups of water molecules in the boundary layer of a nanobubble, which can cause a contraction of its walls and partial collapse of its cavity. However, if the H_3_O fragment appears directly at the boundary of a nanobubble, its pyramidal configuration will lead to the deformation of the nanobubble surface slightly inward. This, in turn, can cause the local instability of the hydrogen-bond network. In this case, the migration of an excess proton along the hydrogen-bond network from the boundary molecule (H_3_O fragment) into the aqueous phase will be accompanied by the elimination of the local stress of the network without the formation of unbound OH groups. Consequently, the localization of hydronium ions should be more stable at a distance of no less than two molecular layers from the water–nanobubble boundary rather than directly at the interface.

Thus, we can claim that in spherical boundary layers (approximately trimolecular in thickness) of nanobubbles in the bulk water, which is in equilibrium with the atmosphere, bicarbonate and hydronium ions, which appeared as a result of the hydration of carbon dioxide molecules, can be steadily localized. Bicarbonate ions can be embedded in water molecular layers by orienting the OH bonds toward themselves and, hence, reducing the number of OH groups, which are not involved in the H-bond network and directed inside the cavity. Hydronium ions, being also localized near the nanobubble–water interface, locally promote the orientation of the OH groups of water molecules away from the cavity. In this case, the higher mobility of hydronium ions in water provides their more uniform distribution in the system, whereas less mobile bicarbonate ions (as locally reinforcing elements of the hydrogen–bond network) should be essentially localized. The thickness of the boundary region between the gas core of a nanobubble and the bulk water, in accordance with the discovered structural features of the clusters that involve bicarbonate and hydronium ions, should be about 1 nm. The charge of the nanobubble is expected to be localized chiefly within this boundary spherical region, and this charge should be negative due to the aforementioned character of the localization and mobility of ions.

These expected structural and energetic features of nanobubbles can form the basis for constructing a phenomenological model of the species and describing their dynamic characteristics, which particularly predetermine the presumable generation of electromagnetic waves.

## 3. Discussion

### 3.1. Stability of Nanobubbles: Bubbles Stabilized by Ions

In terms of the macroscopic phenomenological theory, the key problem of the long-term existence of gas nanobubbles under normal conditions, i.e., far from the boiling point, is the difference in the content of dissolved gases inside nanobubbles and in the surrounding layers of water. Indeed, as follows from the Laplace Equation*P_in_* = *P*_0_ + 2Г/*R_b_*(2)
(where *P_in_* is the gas pressure inside the bubble, *P*_0_ is the atmospheric pressure, Г is the surface tension factor, and *R_b_* is the bubble radius), at small *R_b_* radii the gas pressure inside a bubble will be higher than the atmospheric pressure. Insofar as the content of gas dissolved in a liquid is determined by its partial pressure in the atmosphere, the solution of gas in the liquid will not be saturated with respect to the pressure of the same gas inside the bubble. Such a bubble will not be diffusionally stable; the gas should escape from this bubble into the bulk liquid, and the bubble should collapse [37].

The lifetime of such a bubble is proportional to *R_b_*^2^ [38]. As was found in the paper, at *R_b_* = 100 nm the lifetime of such a bubble is 0.1 ms. In a recent paper [39], it was shown that the lifetime of gas microbubbles increases in aqueous electrolyte solutions with an increase in the ionic concentration. However, for mechanical and diffusion equilibrium of such bubbles, the surface tension should be compensated somehow. Quite a few papers have been devoted to the study of various mechanisms of stabilization of gas nanobubbles; see, for example, review [40] and papers [41,42,43,44,45,46].

A possible mechanism of the surface tension compensation is the localization of structure-breaking (chaotropic) anions in the surface layer of the gas bubble. Structure-breaking anions can steadily exist inside an asymmetric solvation shell, which is monomolecular or even incompletely formed on one side [47,48,49,50]. In [51], the possibility of localizing structure-making (kosmotropic) anions on the bubble surface was theoretically considered as well, and a number of experiments were described in which the stabilization of gas nanobubbles provided by them was confirmed. It is assumed that negatively charged ions stretch the bubble, counterbalancing the surface tension forces. The presence of anions is confirmed by the fact that gas bubbles move toward the anode during electrophoresis, i.e., the ζ-potential at the slipping plane of the bubbles is negative [52]. The surface region of a nanobubble, in which the anions are localized, has a nanometer size [53].

The theoretical estimates we have obtained for the bicarbonate anion are in complete agreement with this phenomenological concept. Even a layer of a trimolecular thickness is sufficient for a stable localization of HCO_3_^−^ anions. Thus, a negatively charged spherical layer is formed over the surface of the gas bubble, and an electrostatic component of the Helmholtz free energy appears (the Equations below are given in the CGSE system).(3)$$\Phi_{e}\left(r\right)=\frac{1}{2\varepsilon }\int_{R_{b}}^{r}\left[Q_{0}^{2}\left(x\right)/x^{2}\right]dx,$$
where *Q*_0_ is the charge of the bubble surface of *R_b_* radius, caused by the embedded anions, and *ε* = 82 is the permittivity of water. For a radius of about 50 or 100 nm, the trimolecular surface layer surrounding the gas core of a nanobubble and having its own thickness smaller than 1 nm can be conventionally referred to as the surface of the nanobubble. In this case, the specific free energy related to the electrostatic component is equal to the negative ponderomotive pressure $P_{e}=-{\left(\frac{\partial \Phi_{e}}{\partial V}\right)}_{T},$ which stretches the spherical bubble. Here, $V=\frac{4\pi }{3}\left(r^{3}-R_{b}^{3}\right)$ is the volume of the outer (relative to the gas core) spherical layer of the bubble. In this case, the Laplace Equation (2) can be rewritten as(4)$$P_{in}+P_{e}=P_{0}+2\Gamma /R_{b}.$$

Under certain conditions, the $2\Gamma /R_{b}=P_{e}$ equality will be met, and such a bubble will be stable both mechanically (equality of pressures at the inner and outer surfaces of the bubble) and diffusionally (the gas pressure inside the bubble is equal to the atmospheric pressure). We named such a bubble a bubston, an abbreviation for *bubble stabilized by ions*. The theoretical model [34,54] predicts the radius of a bubston as(5)$$R_{b}=\frac{Q_{0}^{2/3}}{{\left(16\pi \varepsilon \Gamma \right)}^{1/3}}.$$

In pure water free of foreign impurities, which is brought in contact with the atmospheric air for a long time and has a pH = 5.5 due to the dissolution of carbon dioxide and reaction (1), the volume number density of bicarbonate ions can roughly be estimated as *N_i_*~10^17^ cm^−3^. The participation of bicarbonate and carbonate anions (HCO_3_^−^ and CO_3_^2−^) in the formation of nanobubbles is also evidenced by the results of our experiments on the excitation of stimulated hyper-Raman scattering of light in the field of intense optical pumping in degassed water (containing no nanobubbles) and in water with nanobubbles [55,56].

The content of atmospheric gases dissolved in water is determined by the N_L_·K product, where N_L_ = 2.7 × 10^19^ cm^−3^ is the Loschmidt number, K is the gas solubility constant in water under normal conditions, K = 0.02. Thus, the equilibrium content of air components dissolved in water is of an order of 10^17^ cm^−3^, and the liquid itself is saturated with dissolved gas according to diffusion kinetics. In [34], DLS experiments, in which the dependence of the volume number density of bubstons *n_b_* on the concentration of ions in aqueous NaCl solutions under normal conditions was studied, are described. It was found that in liquids with an ionic concentration of 10^−4^–10^−3^ M, the *n_b_* value falls in a range of 10^6^–10^7^ cm^−3^. In the same work, it was shown that if a cell with initially degassed water is opened to the atmosphere, the *n_b_* volume density increases according to the diffusion kinetics with a characteristic time *t* = *h*^2^/*D*, where *h* is the height of the liquid layer in the cell, and *D* is the diffusion coefficient of gas in the liquid. The increase in the volume number density *n_b_* can be accelerated by effectively agitating the liquid sample with the use of, for example, a magnetic stirrer.

It was shown that bubstons are generated as a result of local breakages in the liquid continuum when vortex motions are originated in liquid layers. A liquid layer adjacent to the wall or bottom of a vessel is motionless, whereas the next neighboring layer is involved in the vortex motion, which can lead to the breakage of the continuity of the liquid sample. This means that for an increase in the volume number density of bubstons, it is necessary to actively stir/shake the liquid sample. It is also worth noting that there are many technologies aimed at significantly increasing the volume density of gas nanobubbles. These technologies are generally referred to as bubbling and are based on high-speed vortex circulation of a water–gas mixture, as well as intense mixing and hydrodynamic cavitation [57,58]. It should also be noted that for a reliable measurement of the ζ-potential of gas nanobubbles, it is necessary to increase their volume density. In [59], where, apparently, the ζ-potential of gas nanobubbles was measured for the first time, the spiral-liquid generation technique [58,60,61] and the technique of cyclic pressure increase–decrease [62] were used to enhance their volume number density.

The solution to the combined problem of the bubston radius *R_b_* and its total charge *Q*_0_ within the thermodynamic description yields *Q*_0_~10^4^
*e* = 1.6 × 10^−15^ C and *R_b_*~100 nm [54]. These estimates of the bubston radius were confirmed in previous experiments using DLS and phase microscopy [34,51,63,64]. As follows from the data presented in Figure 1, the most probable radius of nanobubbles after intense vibrational treatment is close to 75 nm, with a width of the peak of their size distribution at half-maximum of about 70–100 nm, depending on the shaking intensity used. Note that, in general, this corresponds to an area in the spherical layer of about 650–700 Å^2^ per one bicarbonate ion, whose charge is not counterbalanced by the positive charge of hydronium ions. This is also consistent with the pH of water saturated with carbon dioxide under atmospheric conditions, provided that, after shaking, most of the bicarbonate ions are localized in the surface layers of the formed nanobubbles. Note also that, as follows from Figure 1, the size of nanobubbles grows with increasing frequency *ν* and force *F* of shaking, i.e., with increasing the power *W*~*νF* of mechanical treatment of the liquid sample.

Note that in the DLS experiments, the so-called hydrodynamic radius *a_c_* > *R_b_* is measured, i.e., it is impossible to measure the bubston radius *R_b_* with high accuracy in these experiments. Indeed, let us consider water in which carbon dioxide is dissolved. The anions are incorporated into the surface layer of bubstons, creating a negative electric charge *Q*_0_ of the surfaces. A screening double electric layer is formed around the negatively charged shell of the gas core of the bubston [65,66]. In a conventional theoretical model of double layers, a dense part, the so-called “Stern layer” with a thickness on the order of several Ångströms (see [65]) near the charged surface, and a diffuse part, the charge distribution and potential in which can be determined in terms of the Poisson–Boltzmann approach, are distinguished (for more details, see [66]). The *Q*_0_ charge creates, in the liquid, which environs the bubston, an electric field close to spherically symmetrical with a potential (*r* is the distance from the center of the bubble, *r* ≥ *R_b_*) that meets the following conditions:$$\varphi \left(\infty \right)=0;\;-\frac{d\varphi \left(R_{b}\right)}{dr}=E\left(R_{b}\right)=Q_{0}/\left(\varepsilon R_{b}^{2}\right)$$

The resulting electric field acts on the dissolved ions, both anions and cations (counterions). As a result, in the solution layer adjacent to the bubston surface, a concentration gradient of anions $n_{i}\left(r\right)$ and counterions $\bar{n_{i}}\left(r\right)$ is created in accordance with the Boltzmann distribution (here and below, the temperature *T* is given in energy units, i.e., the Boltzmann constant *k* = 1):(6)$$n_{i}(r)=n_{i}^{s}\exp\left[-e\varphi \left(r\right)/T\right];\bar{n_{i}}(r)=n_{i}^{s}\exp\left[e\varphi \left(r\right)/T\right].$$

In Equation (6), $n_{i}^{s}$ is the density of ions located far from bubston surfaces, i.e., ions that are not embedded in the spherical shells of nanobubbles and do not participate in the formation of the double layers. Let us assume that, at *r* = *R_b_*, the condition $e\varphi \left(R_{b}\right)/T<<1$ is met. In this case, the Debye–Hückel approximation is often used, which provides linearization of this Equation and an analytical study of the double-layer structure [64]. In this case, the density of the electric charge in the solution $\rho \left(r\right)=e[n_{i}\left(r\right)-{\bar{n}}_{i}\left(r\right)]$, according to (6), has the form $\rho \left(r\right)=-2n_{i}^{s}e^{2}\varphi \left(r\right)/T,$ and the Poisson–Boltzmann Equation becomes linear:(7)$$\frac{d^{2}\varphi \left(r\right)}{dr^{2}}=-\left(4\pi /\varepsilon \right)\rho \left(r\right)=\kappa^{2}\varphi \left(r\right).$$

Here, $\kappa ={\left(8\pi l_{B}\cdot n_{i}^{s}\right)}^{1/2};$
*l_B_ = e*^2^*/*(*εT*) = 7.0 Å is the so-called Bjerrum length.

In this case, the distributions of the $\varphi \left(r\right)$ potential and the *ρ*(*r*) electric charge density in the diffuse part of the double electric layer have the following form:(8)$$\varphi \left(r\right)=\frac{Q}{\varepsilon \left(1+\kappa R_{b}\right)}\frac{e^{-\kappa \left(r-R_{b}\right)}}{r},$$(9)$$\rho \left(r\right)=-\frac{\kappa^{2}Q}{4\pi \left(1+\kappa R_{b}\right)}\frac{e^{-\kappa \left(r-R_{b}\right)}}{r},r\geq R_{b}.$$

In the analysis presented here, the dense part of the double layer (the Stern layer) was not considered. In this case, the ions in the dense part are considered immobile [65], while the ions in the diffuse part are mobile. Let us consider a compound species that includes a gas core of a bubston surrounded by a negatively charged spherical shell with a total radius *R_b_* and an external Stern layer containing counterions. When moving in a viscous liquid, the peripheral ion layers of the diffuse cloud are effectively displaced. The spherical surface of the compound species with an *a_c_* radius, inside which the charge *Q_c_* is localized, separates it from the moving external environment. Such a surface is the so-called slipping surface, and the *a_c_* radius is the hydrodynamic radius. On this surface, the potential is essentially the electrokinetic or ζ-potential; ζ = *Q_c_*/(*εa_c_*). The value of the ζ-potential can be determined in DLS experiments. In [59], the ζ-potential of bubbles with a radius of *R* = *a_c_* = 100 nm was measured, and the value obtained was ζ = −10 mV. From this, we obtain *Q_c_* = 10^−17^ C ≈ 60 *e*, where *e* is the elementary charge.

In general, these measurements do not contradict our estimates. The proper size of a bubston, including only a gas core and a nanometer layer of water molecules around it, containing mainly anions (about 75 nm), should be smaller than the size of a compound species, which additionally involves a layer of water with counterions (of thickness about 100 nm). Hence, the thickness of the Stern layer can be estimated in a first approximation as 25 nm, and the counterions present in it significantly screen the *Q*_0_ charge (see the comments to Equation (3)). Thus, it can be stated that the structure of nanobubbles, based on the above phenomenological model in combination with experimental data and the results of quantum chemical calculations, is internally consistent and can be used for the subsequent analysis.

### 3.2. Possible Electromagnetic Radiation of Bubble Compound Species

As noted in a classic work [67], “water at room temperature under ordinary conditions (open to the atmosphere) is a nonequilibrium open system that exchanges heat and gases with the environment,” and the formation and transformation of gas nanobubbles are also very significant components of such a nonequilibrium state. The nanobubbles are directly involved in the exchange of matter (gas components) and heat with the environment. The existence of flows (gradients) of liquid and heat produces forces that cause deformation of nanobubbles. If the deformation amplitudes are large, such deformations can be irreversible and result in the collapse of nanobubbles, whereas deformations of small amplitudes are reversible and can cause oscillations of nanobubbles. Oscillations of a charged species, such as a nanobubble, should be accompanied by the absorption/emission of electromagnetic waves at the inherent frequency.

There exist two mutually complementary models of oscillating compound species: (1) oscillations promoted by a periodic external force and (2) oscillations caused by the thermal motion of particles in the surrounding environment.

#### 3.2.1. Model 1: Oscillations Promoted by a Periodic External Force

Let us consider oscillations of a compound species with an initial radius *a_c_* = *R*_0_ (the radius *R* changes with time during oscillations, *R* = *f*(*t*)) and a charge *Q_c_ = Q* = 60 *e* = 10^−17^ C. The analysis below is based on the approach described in [68], namely, the Rayleigh–Plessett Equation for oscillations of a gas bubble in an acoustic pumping field, modified for a charged bubble, is solved:(10)$$\begin{array}{l} \left[\left(1-\frac{\dot{R}}{c_{s}}\right)R+\frac{4\eta }{c_{s}\rho }\right]\ddot{R}=\frac{1}{\rho }\left(P_{0}-P_{v}+\frac{2\Gamma }{R_{0}}-\frac{Q^{2}}{8\pi \varepsilon R_{0}^{4}}\right){\left(\frac{R_{0}}{R}\right)}^{3\gamma }\left(1+\frac{\dot{R}}{c_{s}}\left(1-3\gamma \right)\right)-\frac{{\dot{R}}^{2}}{2}\left(3-\frac{\dot{R}}{c_{s}}\right)+\\ \frac{Q^{2}}{8\pi \varepsilon R^{4}}\left(1-\frac{3\dot{R}}{c_{s}}\right)-\frac{2\Gamma }{\rho R}-\frac{4\eta }{\rho }\left(\frac{\dot{R}}{R}\right)-\frac{1}{\rho }\left(P_{0}-P_{v}+P_{s}\sin\left(\omega t\right)\right)\left(1+\frac{\dot{R}}{c_{s}}\right)-\frac{R}{\rho c_{s}}P_{s}\omega \cos\left(\omega t\right). \end{array}$$
where Г is the coefficient of surface tension, γ = 5/3 is the adiabatic index, *c_s_* is the speed of sound in water, ρ is the density of the liquid, η is the dynamic viscosity, *P*_0_ = 101 kPa is the atmospheric pressure, *P_v_* = 2.34 kPa is the pressure of water vapor at room temperature, *P_s_* and ω are the amplitude and frequency of the sound wave acting on the bubble (compound species), respectively. In this case, in the phases of the sound wave that correspond to reduced pressure, an increase in the radius of a bubble *R* is observed; the growth of *R* occurs at a frequency of ω/2, i.e., precisely in phase with reduced pressure. At the end of this phase, the bubble begins to oscillate at its own frequency.(11)$$\omega_{0}^{2}=\frac{1}{\rho R_{0}^{2}\left(1+\frac{4\eta }{c_{s}\rho R_{0}}\right)}\left(5\left(P_{0}-P_{v}\right)+\frac{8\Gamma }{R_{0}}-\frac{Q^{2}}{8\pi \varepsilon R_{0}^{4}}\right).$$

The attenuation coefficient of such oscillations is expressed as(12)$$\beta =\frac{1}{\rho c_{s}R_{0}\left(1+\frac{4\eta }{c_{s}\rho R_{0}}\right)}\left(4\left(P_{0}-P_{v}\right)+\frac{8\Gamma }{R_{0}}+\frac{4\eta c_{s}}{R_{0}}-\frac{Q^{2}}{8\pi \varepsilon R_{0}^{4}}\right).$$

Note that Equation (10) implies only radial oscillations of gas bubbles, i.e., the bubbles retain their spherical shape. According to Equations (11) and (12), ω_0_ ≈ 10^9^ Hz, and the decay time of such oscillations τ_0_ = 1/β ≈ 4 μs. In this case, as a rule, ω_0_ >> ω, i.e., the action of external pressure at half a frequency *P_s_*sin((ω/2)*t*) is reduced to regular (with a period of 4π/ω) pulses that trigger oscillations of bubbles at their own frequency ω_0_.

When water or an aqueous solution is shaken, its frequency ω typically several hertz (as mentioned above). Based on the analogy with the excitation of own oscillations by low-frequency sound pulses, it can be assumed that shaking a liquid sample can also promote oscillations of bubbles at their own frequencies. Generally speaking, in order to trigger oscillations of nanobubbles at their natural frequency, it is not at all necessary to resonantly swing the bubbles at their natural frequency. Resonant excitation of bubston oscillations at their natural frequency was first produced in our work [69] when stimulated Mandelstam–Brillouin scattering was excited in water under broadband UV pumping. In this experiment, bubston oscillations were triggered by a resonant hypersonic wave, which was excited due to the electrostrictive interaction of the light wave of the pump radiation and the backward Stokes wave.

#### 3.2.2. Model 2: Oscillations Caused by Thermal Motion of Particles in the Environment

In the problem of radiation of a negatively charged bubston, which oscillates at the *ω*_0_ frequency, being surrounded by a polymolecular Stern layer, which contains counterions, we deal with a two-layer system. These are water with a density of 1 g/cm^3^ and a permittivity of ε_1_(ω_0_) ≈ 82 (outer layer) and a gas environment with a density of 10^−3^ g/cm^3^ and a permittivity of ε_2_ = 1 (inner layer). In our case, the radius of the sphere filled with gas is *R* = 75 nm. The electromagnetic wave itself is emitted by ions localized in the spherical layer surrounding the gas core, as well as by ions inside the Stern layer, so that the thickness of the emitting region is on the order of nanometers. We failed to find any example of solutions to the problem of electromagnetic wave emission at the oscillations of charged nanobubbles in the literature. At the same time, there are works where the problem of electromagnetic wave emission by an oscillating charged drop in a gas environment has been solved theoretically; see, for example, review [70].

In solving the problem of excitation of electromagnetic waves by charged compound species, we used the results of theoretical work [71], where the emission of electromagnetic waves by a charged drop in the atmosphere was considered. In work [71], it is assumed that a charged drop has a certain electrical conductivity, i.e., in the bulk of this drop, the electromagnetic wave is screened. At the same time, inside the gas cavity of the bubston, electromagnetic waves emitted by the charged surface should exist. However, in the general case, they should be mutually leveled off due to the existence of counter-propagating components of electromagnetic waves. Accordingly, we take into account only radiation emission into the liquid that environs the compound species, i.e., we consider a two-layer system “liquid-charged superficial layer” rather than a three-layer one. In this approximation, the boundary conditions for radiation on the surface of a charged compound species and a charged drop can be considered the same; the correctness of such an assumption undoubtedly requires additional verification. The analysis given below largely repeats the calculations made in [71] (taking into account the differences between the two-layer and three-layer models), so here we restrict the discussion to presenting the main results.

Note that in the model considered in [71], the damping of oscillations of charged droplets is caused by the emission of electromagnetic waves related to these oscillations. Obviously, this is the only mechanism for the damping of oscillations of charged droplets in a gas environment. However, in our case, oscillations of charged compound species occur in a viscous medium. The damping time of these oscillations is τ_0_ = 1/β ≈ 4 μs (see comments to Equation (11)). In addition, the oscillations of charged droplets arise due to the Rayleigh instability of such droplets in accordance with the condition $\frac{Q^{2}}{16\pi \Gamma R^{3}}\geq 1,$ (where *Q* is the charge of the droplet, *R* is its radius, and Г is the surface tension coefficient) that should be met. Oscillations of a droplet increase due to noises caused by the thermal motion of molecules in the liquid. The motion produces a small-amplitude capillary wave motion on the droplet surface with a characteristic ridge height ξ~(*T*/Г)^1/2^ (see [72,73,74]), where *T* is the temperature in energy units (as before, the Boltzmann constant is considered equal to unity). Therefore, it can be stated that the surface of a charged droplet is unstable, and straining oscillations of such a droplet always exist as long as there is a charge on its surface. In our case, for the onset of oscillations, a redistribution of the volume density of ions in the diffuse layer is necessary. The redistribution can be caused both by the accelerated motion of the compound species during vibrational treatment of the sample. It can also be caused by the differences in the mobility and extents of binding of hydronium and bicarbonate ions in the spherical boundary layer of a nanobubble.

Similar to the model presented in [71], we solved a system of equations that included the wave equation, the Poisson Equation, and the Equation for the fluid velocity field at the boundary of a charged compound species, taking into account the electrostrictive pressure at this boundary. We approximated the electrostrictive pressure on the surface of a compound species as $P_{str}\left(r\right)=\frac{\varepsilon }{8\pi }\left(E^{2}\left(r\right)\right),$
where *r* = *R* + ξ(θ, *t*), *R* is the radius of an unperturbed compound species. By analogy with [71], we sought the solution for the perturbation ξ(θ, *t*) of the spherical surface of the compound species in the form of a series in Legendre polynomials *P_n_*(cos θ), where *n* is the polynomial number, θ is the polar angle:(13)$$\xi \left(\theta ,t\right)=\sum_{n=2}^{\infty }\alpha_{n}P_{n}\left(\cos\theta \right)\exp\left(-i\omega_{n}t\right).$$
where α*_n_* is the initial amplitude of the *n*th oscillation mode, α*_n_*~ξ(θ). In this case, the surface oscillation frequency of a compound species ω*_n_* has real and imaginary parts, which are taken into account in the Equation for the intensity of the radiation of the compound species. For the own frequency ω*_n_*_0_ of the oscillations of a charged sphere of radius R, the equation:(14)$$\omega_{n0}^{2}=\frac{n(n+1)(n-1)}{\rho_{l}n+\rho_{g}(n+1)}\left(\frac{Q^{2}}{4\pi \varepsilon_{l}R^{2}}+\frac{\Gamma (n+2)}{R^{3}}\right)$$
was obtained [71]. Here, Г = 73 dyn/cm is the surface tension coefficient at the gas–water interface, ρ*_l_* = 1 g/cm^3^ is the density of the liquid phase around the nanobubble (water), ρ*_g_* is the density of the gas phase inside the nanobubble, *Q* is the charge of the nanobubble, and *R* = *R*_0_ = 75 nm. As follows from (14), the oscillation frequency of a charged bubble is actually determined by the frequency of the natural oscillations of an uncharged sphere, and the existence of a charge introduces only a slight perturbation in this value. Therefore, the frequency can be estimated as(15)$$\omega_{n0}^{2}=\frac{\Gamma }{R^{3}}\frac{\left(n+1)(n-1)(n+2\right)}{\rho_{l}}$$
and a non-zero contribution appears at *n* ≥ 2, when the excited oscillations transform the sphere into an ellipsoid. At *n* = 2 we obtain ω*_n_*_0_ = 4.2 × 10^9^ Hz.

This frequency is close to ω_0_ = 10^9^ Hz (see Equation (11)), i.e., the natural frequency of the oscillations of a compound species calculated from Equation (14) for *n* = 2 has the same order as the natural frequency of the compound species oscillations ω_0_, which were promoted by the Rayleigh instability; see Equations (10) and (11). Recall that the Rayleigh–Plesset Equation (10) considers only radial oscillations of compound species, i.e., we cannot use the results of integration of this equation to describe an oscillating compound species that preserves its full volume. At the same time, the proximity of the ω*_n_*_0_ and ω_0_ frequencies allows us to assume that the oscillations of the compound species, at which it expands/contracts, and the oscillations, at which its full volume is preserved, but ellipsoidal deformation occurs, lie in the same frequency range, about 1–4 GHz. Note that this spectral interval falls in the frequency range in which the radiation of water samples and aqueous solutions was experimentally recorded (see Figure 2 and the discussion above).

Next, to estimate the intensity of the compound species radiation at the frequencies defined above, we use the solution given in [71], taking into account the additional assumptions made above. In this case, the Equation for the intensity of the radiation of a single compound species with charge *Q* looks as follows:(16)$$I=\frac{Q^{2}}{4\varepsilon (\omega_{n0})}\frac{\Gamma^{n+1}}{c^{2n+1}R^{n+5}}\beta_{n}^{2}{\left[\frac{1}{(2n-1)!!}\right]}^{2}\frac{{\left(n+1\right)}^{n+3}{\left(\left(n-1)(n+2\right)\right)}^{n+1}\rho_{g}}{\rho_{l}^{n+2}n^{2}(2n+1)}$$
where $c=\frac{c_{0}}{\sqrt{\varepsilon }},$
*c*_0_ is the velocity of light in vacuum, ρ*_l_* = 1 g/cm^3^ is the density of water, *R* = *R*_0_ is the radius of a nanobubble, *R* = 75 nm. The coefficient β*_n_*= Δ*R/R*_0_~0.01 is the ratio of the Δ*R* deformation of a compound species to its unperturbed radius *R*_0_, taking into account the invariance of the total volume of the compound species. If we further use *n* = 2, ω*_n_*_0_ = 10^9^ Hz, and ε(ω*_n_*_0_) = 82 in Equation (16), we come to a value *I* ≈ 2.9 × 10^−31^ W/m^2^ for the radiation intensity emitted by one compound species with a charge *Q* ≈ 60 *e* and a radius *R* = 75 nm.

This approximation corresponds to the situation when the objects identified in the DLS experiments are assumed to be spherical nanobubbles of the corresponding radius. However, taking into account that with an increase in the shaking intensity, both the number and the size of nanobubbles increase (see comments to Figure 2), the following situation is also possible: a fraction of nanobubbles of smaller diameter can coalesce to form visible objects of a larger radius. In the latter case, oscillations of smaller nanobubbles should be considered. When a bubble radius decreases by half, its charge (at the same density) decreases by three quarters. Accordingly, the *Q*^2^/*R*^7^ ratio in Equation (16) increases by a factor of 2^3^, which corresponds to an intensity of *I* ≈ 2.3 × 10^−30^ W/m^2^ (note that the frequency of such oscillations will also be different, namely, nearly thrice as high). Coalescence of nanobubbles becomes more probable at their ellipsoidal deformations (which accompany oscillations): an ensemble of coupled polarized ellipsoids arises, which, due to the existence of electrostatic forces, predetermines synchronous (resonant) oscillations of all nanobubbles. The *A*_0_ amplitude of oscillations of the electric field intensity of a coherent ensemble of *k* oscillators is *A*_0_ = *k*·*A*, where *A*^2^ = *I* ≈ 2.3 × 10^−30^ W/m^2^. Then the radiation intensity of a coherent ensemble of small nanobubbles, the number of which should be around 10 for the considered ratio of linear dimensions (1:2), will be *I*_0_ = *k*^2^*I* ≈ 2.3 × 10^−28^ W/m^2^.

Insofar as it is not known which fraction of the observed objects are individual nanobubbles of larger radius and which are aggregates of smaller nanobubbles, it can only be stated that the obtained values (2.9 × 10^−31^ and 2.3 × 10^−28^ W/m^2^) can be considered as the lower and upper estimates of the intensity of radiation emitted by nanobubbles in water. These theoretical estimates agree with the experimental results presented in Figure 2.

## 4. Materials and Methods

### 4.1. Sample Preparation

For studying the effect of shaking on the volume number density and size of nanobubbles, we prepared samples of water and aqueous NaCl solution with a concentration of 10 mg/L, which were shaken at a frequency of 4 Hz with a force of 4 or 8 N (kg × m/s^2^). The frequency and intensity of shaking were controlled with a custom-made Dynamizer device using a Tenzometry Unit online monitoring software ver. 3.1.1 developed based on the Laboratory Virtual Instrumentation Engineering Workbench (LabVIEW 2019, Version 1.4.15.5.). Similar samples, but not subjected to vibrational treatment, served as references. All samples were prepared from water with a resistivity of 18 MΩ × cm at 25 °C purified using a Milli-Q apparatus (Millipore, Merck KGaA, Darmstadt, Germany) and stored in tightly sealed 1000 mL glass jars (Simax^®^, Prague, Czech Republic).

For investigating the ability of samples to emit electromagnetic waves in the radio frequency range, water or NaCl solution was subjected to a series of successive hundredfold dilutions with Milli-Q water. The dilution process was accompanied by intense mechanical shaking at each dilution step at a controlled shaking frequency of 4 Hz and a force of 4 or 8 (kg × m/s^2^). The frequency and intensity of shaking were controlled using a Dynamizer setup with Tenzometry Unit software, ver. 3.1.1. The number of hundredfold dilutions was 12, i.e., the initial concentration should have been reduced formally by a factor of 10^24^. However, in reality, the dilution factor differs greatly from the theoretical estimate due to the flotation of gas nanobubbles, which adsorb molecules of the diluted substance at their interface [75,76]. Samples were prepared in 40 mL vials (Glastechnik Grafenroda, Geratal, Germany) and then poured into 500 mL glass jars (Simax^®^, Prague, Czech Republic), and stored in the tightly sealed jars at room temperature in a dark place.

### 4.2. Dynamic Light Scattering Measurements of the Volume Number Density of Nanobubbles and Water Radiometry

DLS was used to determine the sizes of nanobubbles and their volume number density. Experiments were carried out using a Zetasizer Ultra/Pro setup (Malvern Panalytical Ltd., Malvern, UK). The technique is described in detail elsewhere [77,78]. The DLS setup is equipped with a continuous wave He-Ne laser with a wavelength of λ = 633 nm (maximum power 4 mW) and a temperature controller. The scattering angle was 173°.

We carried out experiments on radiometry of liquid samples using a custom-made unit shown in Figure 8.

A holder with a TES-92 electromagnetic wave detector (TES Electrical Electronic Corp., Taipei., Taiwan) and a thermoshaker (PST-60HL-4, Biosan, Riga, Latvia) for heating a 60 mm polystyrene Petri dish (11000262, FL Medical, Torreglia, Italy), which contained a test sample, were placed inside a Faraday cage, which consists of an aluminum frame covered with a copper mesh with a cell side of 0.56 mm. Measurements were carried out in a spectral range of 50 MHz to 3.5 GHz using the “MAX AVG” mode, which records the maximum intensity level of the electromagnetic radiation. In order to suppress noise and random spikes, the experimental data were averaged over 500 ms (the time constant of the device). First, the experimental samples were poured into a Petri dish and covered with a disposable lid. After that, the sample was heated using the thermoshaker for about 10 min. The surface temperature of the sample was monitored with a contactless laser IR thermometer (235116640, B.Well, Widnau, Switzerland). The radiometer was mounted above the sample when a temperature of 37 °C on the sample surface was reached. The distance between the lower edge of the radiometer and the sample was set to about 0.5 cm, and measurements were carried out for 10 min.

### 4.3. Quantum-Chemical Simulations of Ionic Clusters Within the Surface Layers of Water Molecules

To simulate the state of bicarbonate and hydronium ions in water and clarify their role in the formation of surface layers of molecules around gas cores of nanobubbles, nonempirical quantum chemical calculations were carried out for model cluster systems, which involved water molecules and an additional ion: [HCO_3_(H_2_O)*_n_*]^−^ and [H_3_O(H_2_O)*_n_*]^+^ with the number of water molecules *n* = 20–60.

The model systems of different kinds were considered in Section 2.2. These were the most compact aggregates of molecules stabilized by the largest possible number of hydrogen bonds; aggregates with defects in the hydrogen-bond network; and aggregates, in which the centers of mass of water molecules were located, on average, on the surface of a sphere of a selected radius. Additional ions (bicarbonate and hydronium) were initially placed in different parts of the cluster structures. All structures were optimized, and their correspondence to the minima of the adiabatic potential was confirmed by the normal-coordinate analysis.

For a semi-quantitative characterization of the depth of the local minima of the adiabatic potential (and simultaneously the height of the energy barriers on the paths of structural reorganization of the configurations found), dynamic calculations in the Born-Oppenheimer approximation (Born–Oppenheimer molecular dynamics) were performed. In these calculations, thermal vibrational excitation of previously optimized structures was simulated under normal conditions, for which all vibrations of the cluster with frequencies no higher than 210 cm^−1^ were activated at the initial time moment. The total length of the dynamic trajectories ranged from 5 to 10 ps with a step duration of 0.5 fs, depending on the molecular size of the system. In the case when the analysis of changes in the potential energy of the cluster during its dynamical propagation revealed minima that were comparable in depth to the one that corresponded to the initial configuration, these instantaneous structures were additionally optimized in stationary calculations. From the set of cluster configurations determined in this way, those that had the lowest energies within each of the three above structural kinds (compact, defective, and cavity) were selected.

To estimate thermal contributions to the cluster energies in terms of statistical thermodynamics for the canonical Gibbs ensemble, vibrational contributions to the Gibbs energy of systems (*G_vib_*) under normal conditions (298 K, 1 atm) were estimated. Thus, the model systems mimicked instantaneous fragments of either a virtually unperturbed H-bonded water structure or monomolecular cavity walls that can serve as prototypes of nanobubble boundaries. Optimization of the structures and dynamic calculations were carried out using the density functional method with the B3LYP hybrid exchange-correlation functional [79,80,81,82], which correctly predicts the character of the electron density distribution in water clusters, including those with a nonzero charge, in the presence of both water ions and foreign ions (bicarbonate). One-electron functions were approximated using a double-zeta Gaussian basis set supplemented with polarization functions on all nuclei 6-31G(d,p) (6-31G(d) in the case of O and C, and 6-31G(p) in the case of H) [83,84,85], which is sufficiently flexible for describing large ensembles of particles and at the same time relatively compact to eliminate linear dependence that can lead to artifacts [86].

We have shown previously that the data of such stationary calculations can serve as a basis for estimating both the average structural and energy characteristics of dynamically changing ensembles of particles on 10-ps intervals (which is sufficient to take into account the actual vibrational perturbation of the systems) and the features of the organization of local fragments of macrosystems (including radial distribution functions) provided that the effects produced by foreign particles are local [86]. Nonempirical calculations were carried out with the use of the quantum chemical software package Firefly 8.2 [87], which is partly based on the GAMESS US software code [88]. The Chemcraft graphical package, ver.1.8 [89] was used for analysis and visualization of the results.

### 4.4. Experimental Data Analysis

The data analysis and visualization were performed using R (version 4.0.2, R Foundation for Statistical Computing, Vienna, Austria) and Microsoft Excel (version 2016, build 16.0.2566.1000). The results are shown as descriptive statistics, specifically as the arithmetic mean ± standard deviation (SD). The Shapiro–Wilk test was employed to evaluate the normality of the data distribution, while the Bartlett test was used to assess variance homogeneity. Group comparisons were based on the Student’s *t*-test. Statistical significance was defined as *p* < 0.05.

## 5. Conclusions

The obtained boundary estimates of the radiation intensity of an individual nanobubble are quite small. The concentration of nanobubbles after vigorous shaking is *n_b_* = 10^10^ cm^−3^; see comments to Figure 1. In line with the above consideration, all compound species can be divided into two types: individual larger nanobubbles, the fraction of which is *n*_1_, and coalesced ensembles of smaller nanobubbles, the fraction of which is *n*_2_, where *n*_1_ + *n*_2_ = 1. Objects of each type can be assumed identical in the first approximation. With their total volume concentration *n_b_* ≈ 10^10^ cm^−3^, the average distance between compound species should be about 5 microns. If the initial ellipsoidal deformation of the nanobubbles occurs due to shaking of the sample, then it can be assumed that (with the same sizes, and therefore the same effect exerted by the external vertically directed force) the initial phases of their oscillations coincide. Then we are dealing with a coherent ensemble of two types of radiating oscillators. Depending on the concentrations of particles of each type (*n*_1_ and *n*_2_), the total radiation intensity inside a volume of the sample contained in a Petri dish (about 80 cm^3^) can be estimated as follows:*I*_0_ = (80*n*_1_*n_b_*)^2^*I*_1_ + (80*n*_2_*n_b_*)^2^*I*_2_ ≈ (*n*_1_)^2^ 2.3 × 10^−9^ + (*n*_2_)^2^ ≈ 1.8 × 10^−6^ (W/m^2^).(17)

It is evident that the terms in Equation (17) represent the lower and upper limits of the estimated radiation intensity of a real system in the GHz range. A real liquid phase (water or NaCl solution) is characterized by a size distribution of nanobubbles, which makes it necessary to take into account the contributions of more than two types to the total radiation. Moreover, the oscillations of some objects will inevitably be desynchronized. Nevertheless, the obtained estimate (when compared to the radiometric data; see Figure 2) indicates that, with a high probability, a noticeable part of nanobubble objects in water after shaking are coalesced ensembles of smaller bubbles, whose contribution to the total radiation of the sample should be significant. The estimates can be refined by taking into account the particle size distribution.

From the results of the DLS experiments shown in Figure 1, it can be concluded that the radii of the compound species in water samples subjected to vigorous shaking fall in a range of *R* ∊ (50 – 95) nm. This estimate was made for water subjected to shaking at a frequency of 4 Hz and a force of 8 (kg × m/s^2^) (black curve), at the half-height of the experimental curve. According to Equation (11), this corresponds to the spectral range Δω ≈ 1–1.4 GHz. In order to compare the theoretical estimate (17) to the experimental data shown in Figure 2, an integral estimate of the spectral density of the compound species radiation is required; this will be done in our forthcoming studies.

## Figures and Tables

**Figure 1 ijms-26-06811-f001:**
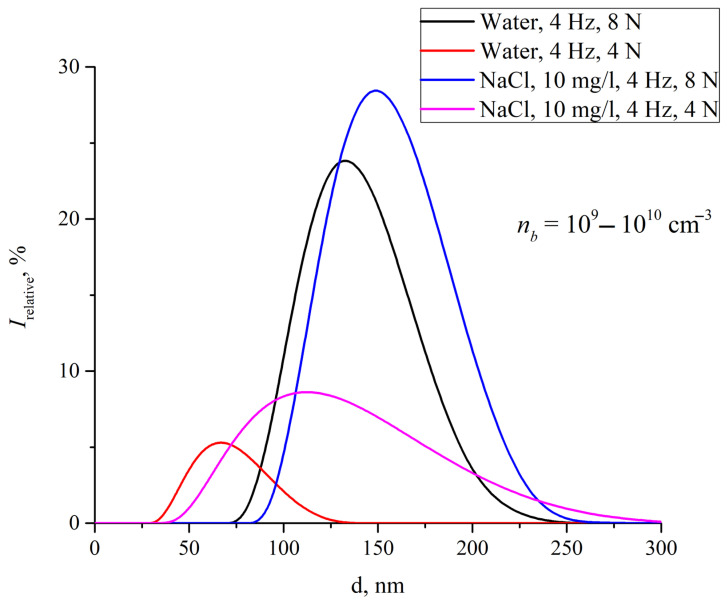
Effect of vibrational treatment of different intensities on the distribution of hydrodynamic diameters of gas nanobubbles in water and NaCl solution, as measured by dynamic light scattering.

**Figure 2 ijms-26-06811-f002:**
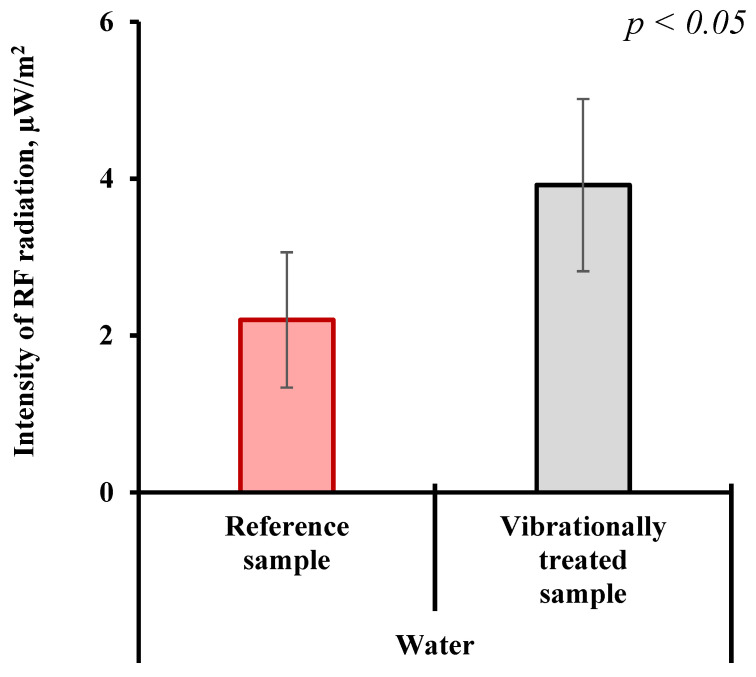
Radiofrequency (RF) emission in the GHz range from water samples subjected to vigorous shaking. The grey column represents to the highly diluted and vibrationally treated water sample. The red column represents to the reference sample (intact water). Sample preparation is described in Section 4.2. Data are presented as arithmetic means ± SD. n = 5, *p* < 0.05 (Student’s *t*-test).

**Figure 3 ijms-26-06811-f003:**
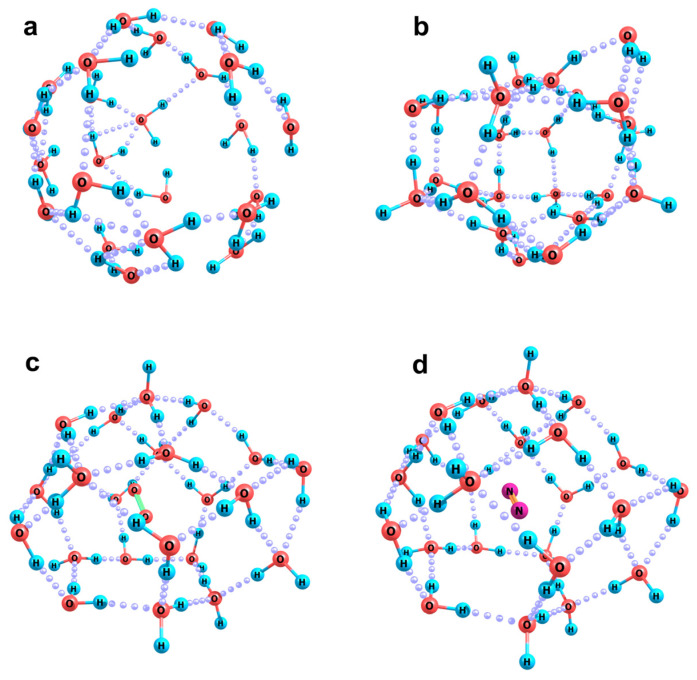
Panel (**a**): random initial; Panel (**b**): optimized arrangement of molecules in a cavity-like (H_2_O)_23_ cluster; and Panels (**c**,**d**): optimal O_2_(H_2_O)_23_ and N_2_(H_2_O)_23_ complex clusters.

**Figure 4 ijms-26-06811-f004:**
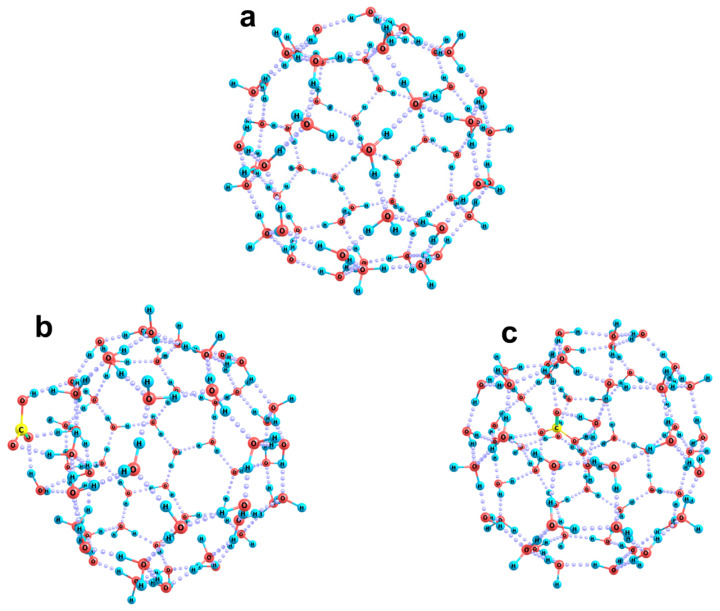
Optimized configurations of (H_2_O)_60,_ Panel (**a**), and [HCO_3_(H_2_O)_60_]^−^. Panels (**b**,**c**), clusters with (**b**) external and (**c**) internal location of the HCO_3_ fragment with respect to the initial water cluster sphere.

**Figure 5 ijms-26-06811-f005:**
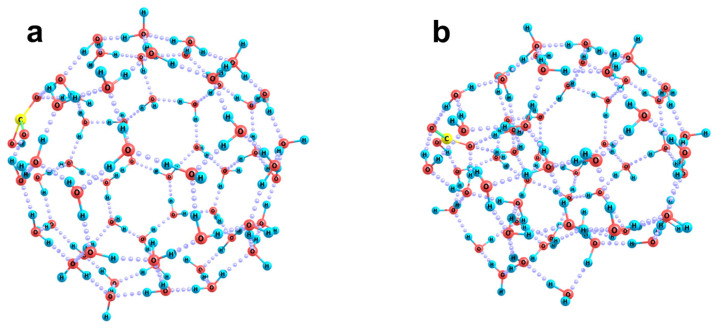
Reorganization of the [HCO_3_(H_2_O)_60_]^−^ cluster upon inclusion of the HCO_3_ fragment into a spherical monolayer of water molecules: initial configuration, Panel (**a**), and the result of steady-state relaxation of the cluster, Panel (**b**).

**Figure 6 ijms-26-06811-f006:**
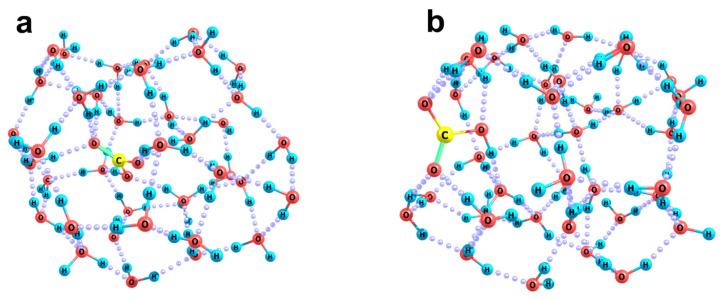
Optimized structures of the ionic cluster [HCO_3_(H_2_O)_39_]^−^ with internal, Panel (**a**), and superficial, Panel (**b**), locations of the bicarbonate ion.

**Figure 7 ijms-26-06811-f007:**
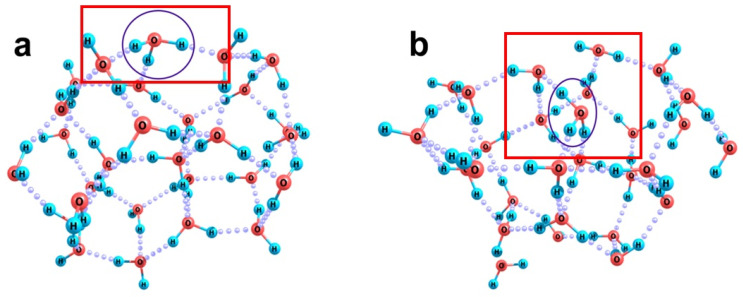
Optimized structures of the [H_3_O(H_2_O)_27_]^+^ cluster with surface, Panel (**a**), and internal, Panel (**b**), positions of the hydronium ion, which is encircled for clarity.

**Figure 8 ijms-26-06811-f008:**
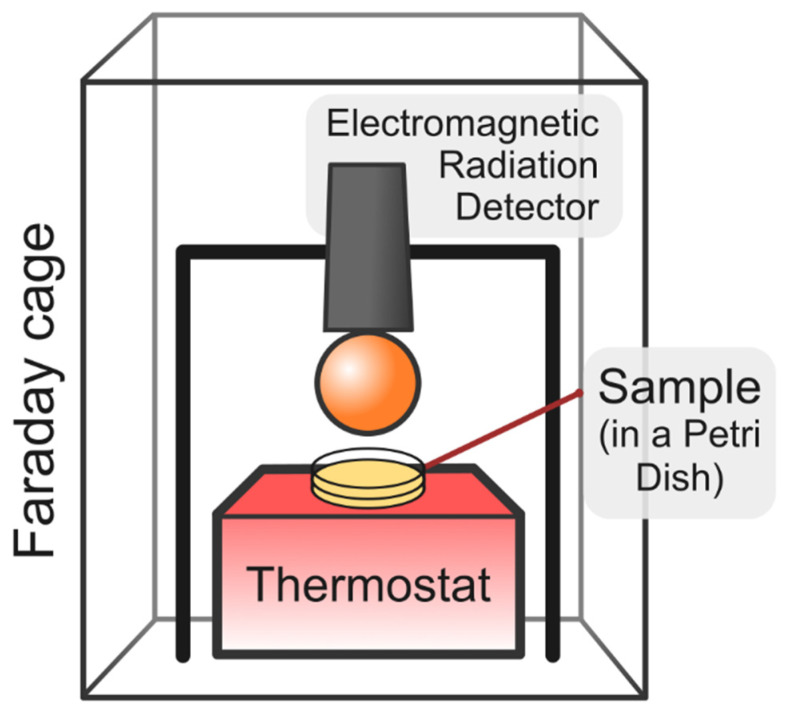
Schematic setup for radiometric measurements.

## Data Availability

The original contributions presented in this study are included in the manuscript. Further inquiries can be directed to the corresponding authors.

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
