# Peer review of "Resonant Oscillations of Ion-Stabilized Nanobubbles in Water as a Possible Source of Electromagnetic Radiation in the Gigahertz Range"

_ijms, 2025, doi:10.3390/ijms26146811_

Round 1

Reviewer 1 Report

Comments and Suggestions for Authors

Minor comments 

1) Axis labels in Fig. 2 (and similar figures) should have units and more descriptive names (e.g., instead of “Power Flux Denisty”).

2) Ensure consistency and correct numbering throughout. For example, citations like [1]–[7] appear grouped frequently, make sure all are accurate and appear in the reference list.  Moreoever, Confirm that all references cited in the introduction (e.g., [15], [16], [31]) are up to date and accessible.

3) Introduce abbreviations like DLS (Dynamic Light Scattering) at their first mention in both abstract and main text. Clearly define all variables in figures and models (e.g., VT in Fig. 2) at first mention.

Comments on the Quality of English Language

It must be improved

Author Response

We are grateful for the reviewer’s time and efforts to evaluate our manuscript. Below, we answered all raised questions.

Question 1: Axis labels in Fig. 2 (and similar figures) should have units and more descriptive names (e.g., instead of “Power Flux Density”).

Answer 1:

We have unified the units on the graph and in the text. We have also changed the designation of the ordinate axis. In the revised version this designation is “Intensity of RF radiation”.

Question 2: Ensure consistency and correct numbering throughout. For example, citations like [1]–[7] appear grouped frequently, make sure all are accurate and appear in the reference list.  Moreoever, Confirm that all references cited in the introduction (e.g., [15], [16], [31]) are up to date and accessible.

Answer 2:

We have completely reviewed and corrected all in-text references. At the moment, they are added via the Zotero reference manager and formatted according to the MDPI style. We hope that there are no more errors in the presentation.

Question 3: Introduce abbreviations like DLS (Dynamic Light Scattering) at their first mention in both abstract and main text. Clearly define all variables in figures and models (e.g., VT in Fig. 2) at first mention.

Answer 3:

We carefully reviewed the manuscript and added abbreviations at the first mention of the term (both in the abstract and in the main text). (Except for commonly used DNA (deoxyribonucleic acid); hertz (megahertz - MHz, gigahertz – GHz) etc.)

Comment 1 (Question 4): Quality of English Language must be improved

Answer 4:

We have revised the entire text for English quality and made many corrections. All corrections are visible in the corrected version as tracked changes.

Reviewer 2 Report

Comments and Suggestions for Authors

Dear Editor,

In their paper titled “Resonant oscillations of ion-stabilized nanobubbles in water as a possible source of electromagnetic radiation in the gigahertz range,” Bunkin et al. proposed a theoretical model for gas nanobubbles and their potential for electromagnetic radiation. The authors initiated their study with experiments to observe the population of nanobubbles, followed by simulations, and ultimately proposed a theoretical model. In my opinion, the paper is interesting and may be considered for publication. However, before further progress can be made, there are several major issues that the authors must address. I have listed my comments below and would be pleased to review the revised version to assist in making a final decision.

  1. Please introduce the symbol N in the abstract before using it without definition.
  2. The novelty of the paper should be clearly stated in the Introduction.
  3. The paper does not follow the conventional structure (i.e., Introduction, Materials and Methods, Results and Discussion, and Conclusion). Instead, it uses a different layout (Introduction, Results, Discussion, Materials and Methods, and Conclusion) which reduces readability and is confusing. It is recommended that the authors adopt the standard format to improve clarity and coherence.
  4. Inside a nanobubble, vapor or non-condensable gases are present. Have the authors considered the effects of these components in their simulations?
  5. Paragraphs should generally consist of 6 to 12 lines. In several instances, the authors use very short, two-line paragraphs. Please revise accordingly for better readability.
  6. The validation section is missing, which is a critical weakness of the paper. Please provide validation criteria to support the obtained results and theoretical model.
  7. Please add the unit for the permittivity of water.
  8. For all equations, it is essential that all variables are clearly defined. Alternatively, the authors should provide a complete Nomenclature table. In several equations, not all variables are defined. I would be happy to review this aspect again in the revised version.
  9. The pressure and concentration of vapor or gas inside the nanobubble are higher than in free conditions. Have the authors accounted for this in their model and simulations?
  10. In reality, nanobubbles can be observed both on surfaces and in free suspension. They may also undergo collapse and shrinkage. The question is: which physical context does this study address? If a nanobubble collapses, how can it be observed, and how can the ions continue to oscillate? For studies on collapsing and surface bubbles, the following papers may be of interest to the authors.

Understanding the stabilization of a bulk nanobubble: A molecular dynamics analysis. Langmuir, 2021.

The role of sawtooth-shaped nano riblets on nanobubble dynamics and collapse-induced erosion near solid boundary. Journal of Molecular Liquids, 2024.

Minimum current for detachment of electrolytic bubbles. Journal of Fluid Mechanics, 2023.

Author Response

Question 1: Please introduce the symbol N in the abstract before using it without definition.

Answer 1:

By the designation "N," we meant the force value in Newtons. In the revised version we replaced it with kg×m/s2. The abbreviation in the form of the letter N remained only in the legend to graph 1, and it is explained in the legend.

Question 2: The novelty of the paper should be clearly stated in the Introduction.

Answer 2:

In the revised version, we have made changes to the Introduction that highlight the novelty of our work.

Corrections in lines 445 - 496: “In summary, this work is devoted to the study of physical mechanisms of electromagnetic wave emission from liquid samples, and these mechanisms are not associated with bioluminescence, i.e., with the emission of biological macromolecules due to chemiluminescence, which is excited, as a rule, in the presence of reactive oxygen species. The emission of electromagnetic waves in the model under consideration is caused by oscillations of electrically charged gas nanobubbles at their eigen-frequency. Such oscillations are accompanied by the emission of an electromagnetic wave.”

Question 3: The paper does not follow the conventional structure (i.e., Introduction, Materials and Methods, Results and Discussion, and Conclusion). Instead, it uses a different layout (Introduction, Results, Discussion, Materials and Methods, and Conclusion) which reduces readability and is confusing. It is recommended that the authors adopt the standard format to improve clarity and coherence.

Answer 3:

Yes, you are right, the structure is different from the usual, but only because the editors require it. According to the submission rules, the Materials and Methods section should be placed closer to the end (between the discussion and conclusions). (https://www.mdpi.com/files/word-templates/ijms-template.dot)

  • Introduction
  • Results
  • Discussion
  • Materials and Methods
  • Conclusions

    Question 4: Inside a nanobubble, vapor or non-condensable gases are present. Have the authors considered the effects of these components in their simulations?

    Answer 4:

    As follows from our theoretical model, the gas pressure inside the bubstons must be equal to the atmospheric pressure, i.e. 10^5 Pa; this is the condition of diffusion stability of the bubstons. We consider the formation of bubstons under normal conditions, i.e. at room temperature. At room temperature, the pressure of saturated water vapor is 2.3x10^3 Pa. Water molecules (if they occasionally appear inside the gas phase) easily and readily attach to the existing water boundary layers. Carbon dioxide molecules are rapidly hydrated, and the state of thus appearing bicarbonate and hydronium ions is specially modeled in the work (see the section devoted to the results of nonempirical simulations). Then, it seems reasonable to consider only molecular nitrogen and oxygen as gases that constitute the gas core of nanobubbles. The role of these molecules was actually considered, but we did not include the data in the original version of the manuscript because we treated them as supplementary aimed at properly selecting cluster structures for subsequent analysis. However, probably, this was not a good idea. It is reasonable to show a reader which effects are promoted by the presence of diatomic gas molecules inside cavities formed by water molecules and, hence, clusters of which kind can serve as reasonable models of nanobubble fragments. In the revised version, these data and their discussion are added on pp. 6-7.

    Corrections in lines 445 - 496:

    To provide the local arrangements of molecules in the model clusters that can be considered as similar to those in nanobubble boundary layers, we have carried out preliminary test simulations. We have optimized the structures of individual water clusters, the molecules in which were initially randomly distributed over the sphere surfaces of a gradually increasing radius. The number of molecules was selected in such a way that the final H-bonded network can envelop the whole sphere. The initial radii considered fell in the range of 3 to 7 Å, with the largest number of molecules being 60. An example of the initial cluster composed of 23 water molecules arranged over a sphere of 4-Å radius is shown in Fig. 3a. During the optimization procedure, no restriction was imposed. Many resulting structures were characterized by continuous 3D H-bond networks with small cavities inside. Only a few configurations could be classified as hollow clusters. The best variant of this kind composed of 23 molecules is shown in Fig. 3b. In the case of the large numbers of molecules (50 to 60) ball-like clusters were found (such as the one shown in Fig. 4a). Then, O2 or N2 molecules were placed inside the hollow clusters, and the resulting complex systems were again optimized. The above 23-molecular cluster was found to be the smallest one, in which the gas molecules can be kept, with the vertical dissociation energy of the systems being close to zero. The energy was estimated taking into account the basis set superposition error as follows:

    De = ½ (Ei(X2)+ Ei((H2O)n)) + ½ (Ef(X2)+ Ef((H2O)n)) - E((X2(H2O)n),

    where E((X2(H2O)n) is the total energy of the cluster at its optimized configuration, and Ei(F) and Ef(F) stand for the energies of the corresponding fragment (F = X2 or (H2O)n and X = O or N) taken at its configuration within the cluster and calculated either with the use of the individual basis set of the fragment (Ei) or with the use of the full basis set of the whole cluster (Ef).

    Fig. 3. (Panel a) Random initial and (Panel b) optimized arrangement of molecules in a cavity-like (H2O)23 cluster and (Panels c, d) optimal O2(H2O)23 and N2(H2O)23 complex cluster.

    At the smaller number of molecules, the De values in the case of N2 and O2 molecules were negative. It is worth noting that when these molecules were placed in the center of the cavity formed by 23 water molecules and the whole system relaxed, the cavity shape changed (Figs. 3c, 3d). It became less distorted and more ball-like. This is reflected in both the surface areas of the clusters and their volumes estimated in terms of the overlapping atomic van-der-Waals spheres (r(H) = 1.2 Å and r(O) = 1.5 Å):

    (H2O)23

    O2(H2O)23

    N2(H2O)23

    Surface area, Å2

    551

    597

    601

    Volume, Å3

    383

    407

    410

    Note that the formal volume per one gas molecule in these clusters is smaller than that in a gas phase under normal conditions by a factor of nearly 200, which corresponds to the actual pressures inside nanobubbles that appear under extreme conditions. Under normal conditions, the pressure should be much lower, which is the case of cavities formed by the larger numbers of water molecules (about 60). Anyhow, diatomic gas molecules contained in H-bonded hollow clusters determine their less defective and more ball-like shapes with no OH groups uninvolved in hydrogen bonds inside the cavity. This is close to actual structures of nanobubbles. Therefore, water clusters of this kind composed of larger numbers of molecules were selected for subsequent analysis of the role of CO2 molecules, whose appearance in water should lead to the formation of bicarbonate and hydronium ions. The state of the ions within the boundary layers of nanobubbles is of primary interest.

    Question 5: Paragraphs should generally consist of 6 to 12 lines. In several instances, the authors use very short, two-line paragraphs. Please revise accordingly for better readability.

    Answer 5:

    The text was revised accordingly.

    Question 6: The validation section is missing, which is a critical weakness of the paper. Please provide validation criteria to support the obtained results and theoretical model.

    Answer 6:

    There is no such special section in the manuscript. However, every time a principal conclusion is formulated based on the consistency of the experimental observations and theoretical results obtained, the attention of the reader is drawn to it throughout the manuscript. We can mention only the following key statements. At first, nonempirical modeling gives us grounds to state that bicarbonate ions (which appear in water as a result of the hydration of carbon dioxide) can steadily be localized within spherical boundary layers (approximately trimolecular, about 1 nm in thickness) of nanobubbles and that these ions are much less mobile compared to hydronium ions, which particularly predetermines the negative charge of nanobubbles. Then, it is said that the charges of bubstons (nanobubbles stabilized by ions) estimated in terms of the proposed phenomenological model agree with the expected content of bicarbonate ions in water under normal conditions. All these aspects are summed up as follows on p. 16: “it can be stated that the structure of nanobubbles, based on the above phenomenological model in combination with experimental data and the results of quantum chemical calculations, is internally consistent and can be used for the subsequent analysis.” Later on, it is stressed that it is the boundary layer oscillations of nanobubbles (related to the changes in the shape or diameter of nanobubbles) that can cause electromagnetic waves (the corresponding β parameter in formula (15) is set to 0.01). Furthermore, attention is drawn to the fact that the inherent frequencies of bubstom oscillations estimated in terms of two different models agree with each other as falling in the same range of 1–4 GHz (see p. 19). This is a subrange of a broader experimental range where the radiation of the samples of water and aqueous solutions (which definitely contain nanobubbles according to the preparation technique) was recorded. Finally, the key characteristic of the systems, namely, the intensity of electromagnetic radiation, found according to model II agrees in the magnitude with the measured values, which seems the best and most reliable confirmation of the validity of the models proposed. The latter aspect is specially given as an extended discussion in the Conclusions section to draw the attention of readers to it. It should also be noted that we originally intended to write a paper describing an experiment on electron beam irradiation of liquid samples. In this experiment, it was shown that the absorption spectrum of liquid samples, caused by solvated electrons in these samples, is controlled by the content of nanobubbles; this was confirmed by the proposed theoretical model. We also conducted an experiment in which a liquid sample not irradiated with electrons is subjected to long-term remote action (incubation) of a sample, which was preliminarily stirred and shaken. It was found that due to incubation the uv absorption spectrum of the unshaken incubated liquid sample, not irradiated with electrons, also changes. In addition, the content of nanobubbles in the bulk of the incubated sample increases. This is an indirect confirmation of the electromagnetic wave emission caused by nanobubbles; this wave stimulates resonant oscillations of nanobubbles in the bulk of the incubated sample, which leads to a change in the absorption spectrum and an increase in the density of nanobubbles in the incubated sample. However, the volume of the resulting manuscript turned out to be too large, and therefore we decided to publish the results on spectroscopy of samples irradiated with high-energy electrons in a separate paper, which will also be submitted to the IJMS journal; this forthcoming article should be considered precisely as an experimental confirmation of the theoretical models presented in the manuscript under review.

    Question 7: Please add the unit for the permittivity of water.

    Answer 7:

    Permittivity is a dimensionless quantity. We used the CGSE system of units. Note that the dimensional quantity "vacuum permittivity" is not included in the CGSE system.

    Question 8: For all equations, it is essential that all variables are clearly defined. Alternatively, the authors should provide a complete Nomenclature table. In several equations, not all variables are defined. I would be happy to review this aspect again in the revised version.

    Answer 8:

    It was fixed throughout the text of the new version.

    Question 9: The pressure and concentration of vapor or gas inside the nanobubble are higher than in free conditions. Have the authors accounted for this in their model and simulations?

    Answer 9:

    As follows from the theoretical model presented in our manuscript, the gas content inside gas nanobubbles is determined by the Loschmidt number, which corresponds to normal conditions: room temperature and atmospheric pressure. If the gas pressure inside a nanobubble is higher than atmospheric, then such a bubble can be in a state of mechanical equilibrium, but the condition of diffusion equilibrium will be violated; see the work of Epstein, P.S.; Plesset, M.S. On the Stability of Gas Bubbles in Liquid-Gas Solutions. The Journal of Chemical Physics 1950, 18, 1505–1509, doi:10.1063/1.1747520. Indeed, liquid samples in contact with atmospheric air are saturated with dissolved air, and its content is determined by atmospheric pressure and the solubility constant. If the gas pressure inside the nanobubble is higher than the atmospheric pressure, then the gas solution will no longer be saturated with respect to the gas pressure inside the nanobubble, i.e. the diffusion equilibrium will no longer be kept at the nanobubble boundary. In this case, the gas from the nanobubble will escape into the bulk of the liquid, and such a nanobubble will eventually collapse, i.e. such a nanobubble will no longer be stable. Since we are considering stable / long-lived bubbles, we assume that the gas inside the nanobubbles is under normal conditions: atmospheric pressure and room temperature. At the same time, we have checked whether the results obtained are valid in the situation when nanobubbles appear in water under extreme conditions when pressures and temperatures inside them are much higher than normal ones. As follows from our tentative estimates, the shapes of model clusters (which are ball-like) are actually provided by the presence of gas molecules inside the clusters; and the higher the pressure inside, the more pronounced the convexity of their boundary surface. The corresponding data are covered by the aforementioned additional discussion (see comment no. 4), which can now be found on pp. 6-7 of the manuscript.

    Corrections in lines 445 - 496:

    (same insertion like in Answer 4)”

    Question 10: In reality, nanobubbles can be observed both on surfaces and in free suspension. They may also undergo collapse and shrinkage. The question is: which physical context does this study address? If a nanobubble collapses, how can it be observed, and how can the ions continue to oscillate? For studies on collapsing and surface bubbles, the following papers may be of interest to the authors.

    • Understanding the stabilization of a bulk nanobubble: A molecular dynamics analysis. Langmuir, 2021.
    • The role of sawtooth-shaped nano riblets on nanobubble dynamics and collapse-induced erosion near solid boundary. Journal of Molecular Liquids, 2024.
    • Minimum current for detachment of electrolytic bubbles. Journal of Fluid Mechanics, 2023.

    Answer 10:

    We are grateful to the reviewer for this comment. We have cited the works suggested by the reviewer in the new version of the manuscript. Indeed, nanobubbles can be stabilized by contact with a hydrophobic surface; see the review by Alheshibri, M.; Qian, J.; Jehannin, M.; Craig, V.S.J. A History of Nanobubbles. Langmuir 2016, 32, 11086–11100, doi:10.1021/acs.langmuir.6b02489. The problem of the existence of stable nanobubbles in suspension in the bulk of a liquid appears to be more difficult compared to the problem of stabilization of nanobubbles on a surface. If a nanobubble in the bulk of a liquid is not diffusionally stable, then the lifetime of such a nanobubble is proportional to the square of its radius; see Ljunggren, S.; Eriksson, J.C. The Lifetime of a Colloid-Sized Gas Bubble in Water and the Cause of the Hydrophobic Attraction. Colloids and Surfaces A: Physicochemical and Engineering Aspects 1997, 129–130, 151–155, doi:10.1016/S0927-7757(97)00033-2. It follows that the lifetime of nanometer-sized non-stabilized gas nanobubbles is on the order of several milliseconds. Therefore, in our work we consider nanobubbles, which are stabilized in the bulk of liquid. Various mechanisms of nanobubble stabilization in the bulk of liquid are considered in the literature. We decided on the mechanism of stabilization due to selective adsorption of ions on the surface of nanobubbles. The choice of this particular stabilization mechanism is due to the fact that the mechanism estimated in the theoretical work of Bunkin, N.F.; Bunkin, F.V. Bubston Structure of Water and Electrolyte Aqueous Solutions. Phys.-Usp. 2016, 59, 846–865, doi:10.3367/UFNe.2016.05.037796 the charge of ions adsorbed on the nanobubble surface is consistent with the experimentally measured zeta potential value.

Round 2

Reviewer 2 Report

Comments and Suggestions for Authors

Dear Editor,

In their revised manuscript titled “Resonant Oscillations of Ion-Stabilized Nanobubbles in Water as a Possible Source of Electromagnetic Radiation in the Gigahertz Range,” Bunkin et al. addressed most of my concerns. However, a few minor issues need to be resolved before the manuscript can be recommended for publication.

  1. It appears that the authors did not consider the effects of the bulk water surrounding the nanobubble. If this is correct, how might the omission of these effects influence the results obtained? Please elaborate on this point.
  2. Please clearly state that this manuscript investigates bulk nanobubbles.

Author Response

1. Referee's comment.

It appears that the authors did not consider the effects of the bulk water surrounding the nanobubble. If this is correct, how might the omission of these effects influence the results obtained? Please elaborate on this point.

Our reply.

We are grateful to the reviewer for this comment. It is the properties of the bulk aqueous ionic solution that allow the nucleation and stabilization of gas nanobubbles. Section 3.1 presents the Laplace equation for a spherical bubble in a bulk liquid. Selective adsorption of anions from the bulk aqueous solution occurs within a thin surface boundary layer (the thickness of which is estimated as a result of nonempirical modeling, see below) of such a bubble. In our theoretical model, the properties of the aqueous solution are taken into account in the form of macroscopic material parameters specific to this liquid. These parameters include the dissolved air content, the content of dissolved ions, temperature, and permittivity. Next, we solve the Poisson–Boltzmann problem on the stability of a nanobubble whose shell contains solvated anions (we call them adsorbable ions, or self-ions), while the nanobubble is surrounded by the next layer, which contains counterions (the dense Stern layer), and a diffuse layer, which contains both counterions and self-ions. Here we have used the traditional approach based on the Debye-Huckel approximation. Thus, the properties of bulk liquid were certainly taken into account at the macroscopic level. In addition, these properties were taken into account at the nanoscale level in the quantum chemical modeling. In Section 2.2, we studied the structural features and relative stability of different fragments of the hydrogen-bond network of water molecules in the presence of both individual and foreign ions. As discussed in this section, the changes in the hydrogen-bond network of water molecules, caused by the presence of hydronium or bicarbonate ions, are local in their nature. The changes in the hydrogen-bond network are observed within three (at most) molecular layers around the ion. All clusters used in the modeling include a sufficient number of water molecules to reproduce these local changes in the hydrogen-bond network. In particular, the reader's attention is drawn to the fact that the bicarbonate ions require a trimolecular shell around the gas core of the nanobubble to be steadily localized there. This conclusion was not obvious a priori and is based on the results of modeling. Thus, the conclusions presented in the manuscript about the structural features manifested in the interaction of ions and the hydrogen-bond network of water are reliable; and they are applicable to the bulk water as well. We also took into account that the absolute values of the interaction energy between the ion and the hydrogen-bond network change if a polarizable continuum around the model cluster is added, but the relative energy values for the ionic clusters where ions are screened from the continuum by a sufficient number of water molecules will not change. Finally, the thermal vibrational excitation of water molecules and the corresponding contributions to the Gibbs energy were explicitly taken into account in our model. All these aspects are specially emphasized in the Discussion and Methodical sections.

2. Referee's comment.

Please clearly state that this manuscript investigates bulk nanobubbles.

Our reply.

We are grateful to the reviewer for this comment. In order to take this comment into account, we have rewritten the Abstract. New version of the Abstract looks like this:

It is well known that aqueous solutions can emit electromagnetic waves in the radio frequency range. However, the physical nature of this process is not yet fully understood. In this work, the possible role of gas nanobubbles formed in the bulk liquid is considered. We develop a theoretical model based on the concept of gas bubbles stabilized by ions, or "bubstons." The role of bicarbonate and hydronium ions in the formation and stabilization of bubstons is explained through the use of quantum chemical simulations. A new model of oscillating bubstons, which takes into account the double electric layer formed around their gas core, is proposed. Theoretical estimates of the frequencies and intensities of oscillations of such compound species are obtained. It was determined that oscillations of negatively charged bubstons can occur in the GHz frequency range, and should be accompanied by the emission of electromagnetic waves. To validate the theoretical assumptions we used dynamic light scattering (DLS) and showed that after subjecting aqueous solutions to vigorous shaking with a force of 4 or 8 N (kg×m/s2) and a frequency of 4-5 Hz, the volume number density of bubstons increased by about 2 orders of magnitude. Radiometric measurements in the frequency range of 50 MHz to 3.5 GHz revealed an increase in the intensity of radiation emitted by water samples upon the vibrational treatment. It is argued that, according to our new theoretical model, this radiation can be caused by oscillating bubstons.